# Strong Tribocatalytic Nitrogen Fixation of Graphite Carbon Nitride g-C_3_N_4_ through Harvesting Friction Energy

**DOI:** 10.3390/nano12121981

**Published:** 2022-06-09

**Authors:** Zheng Wu, Taosheng Xu, Lujie Ruan, Jingfei Guan, Shihua Huang, Xiaoping Dong, Huamei Li, Yanmin Jia

**Affiliations:** 1Xi’an Key Laboratory of Textile Chemical Engineering Auxiliaries, School of Environmental and Chemical Engineering, Xi’an Polytechnic University, Xi’an 710048, China; xutaosheng2021@126.com; 2College of Physics and Electronic Information Engineering, Zhejiang Normal University, Jinhua 321004, China; lujie_ruan@foxmail.com (L.R.); huangshihua@zjnu.cn (S.H.); lihuamei@zjnu.cn (H.L.); 3Key Laboratory of Surface & Interface Science of Polymer Materials of Zhejiang Province, Department of Chemistry, Zhejiang Sci-Tech University, Hangzhou 310018, China; xpdong@zstu.edu.cn; 4School of Science, Xi’an University of Posts and Telecommunications, Xi’an 710121, China

**Keywords:** tribocatalysis, g-C_3_N_4_, N_2_ fixation, ammonia generation, friction

## Abstract

Mechanical energy derived from friction is a kind of clean energy which is ubiquitous in nature. In this research, two-dimensional graphite carbon nitride (g-C_3_N_4_) is successfully applied to the conversion of nitrogen (N_2_) fixation through collecting the mechanical energy generated from the friction between a g-C_3_N_4_ catalyst and a stirring rod. At the stirring speed of 1000 r/min, the tribocatalytic ammonia radical (NH_4_^+^) generation rate of g-C_3_N_4_ can achieve 100.56 μmol·L^−1^·g^−1^·h^−1^ using methanol as a positive charge scavenger, which is 3.91 times higher than that without any scavengers. Meanwhile, ammonia is not generated without a catalyst or contact between the g-C_3_N_4_ catalyst and the stirring rod. The tribocatalytic effect originates from the friction between the g-C_3_N_4_ catalyst and the stirring rod which results in the charges transfer crossing the contact interface, then the positive and negative charges remain on the catalyst and the stirring rod respectively, which can further react with the substance dissolved in the reaction solution to achieve the conversion of N_2_ to ammonia. The effects of number and stirring speed of the rods on the performance of g-C_3_N_4_ tribocatalytic N_2_ fixation are further investigated. This excellent and efficient tribocatalysis can provide a potential avenue towards harvesting the mechanical energy in a natural environment.

## 1. Introduction

Nowadays, due to the immoderate consumption and mining of fossil fuels, energy shortage and environment pollution have become critical issues which are a threat to the survival and development of society [1]. Accordingly, it is necessary to look for renewable and green energy to replace these fossil fuels. Ammonia has been regarded as a green energy source with some advantages such as no carbon dioxide emission, high energy density and convenient transportation [2,3]. Nevertheless, how to perform ammonia production is also a vital issue. At present, various methods have been reported to convert nitrogen (N_2_) to ammonia (NH_3_), such as thermocatalytic reduction, electrocatalytic reduction and photocatalytic reduction [4,5,6]. However, thermocatalytic and electrocatalytic reduction usually require high-pressure or high-temperature operating conditions, which limit the actual application of ammonia production [7,8]. Additionally, photocatalytic reduction of nitrogen can be performed at a mild condition, but low light utilization efficiency would seriously hinder the actual production of ammonia [9]. Therefore, it is crucial to seek mild and highly efficient approaches for nitrogen reduction.

In nature, mechanical energy is widespread distributed energy, which exists in wind, river flows and human movement [10]. If such abundant energy can be successfully harvested for the reduction of nitrogen, it would be meaningful to propel the production of ammonia. Tribocatalysis, which can convert the friction of mechanical energy into the electric energy via persistent friction, has attracted the attention of researchers [11,12,13]. Under friction, when two different kinds of materials contact each other, chemical bonds will form through physical contact on the interface [14]. After being separated, two materials will carry the positive or negative charges respectively, due to breakage of the chemical bonds [15]. Then the free charges generated via the triboelectric process can be further applied to the catalytic reaction such as dye degradation, carbon dioxide or flammable gas production [16,17,18,19,20]. However, to date, there has been no report about harvesting the mechanical energy from friction via tribocatalysis for nitrogen reduction to produce ammonia.

Graphite carbon nitride (g-C_3_N_4_) is an emerging semiconductor material with layered structure similar to graphite [21]. The C and N atoms inside it are arranged alternately through sp^2^ hybridization [22]. g-C_3_N_4_ is a candidate catalyst in several catalytic areas such as the decomposition of organic pollutants [23,24,25], hydrogen evolution [26,27] and carbon dioxide reduction [28,29,30] based on its advantages of being insoluble in water and having a large specific surface area and stable chemical properties. In 2019, Xia et al. achieved the photocatalytic synthesis of ammonia from nitrogen through using a defecting g-C_3_N_4_ catalyst. After 100 min light irradiation, the yield of ammonia can be up to 54 μmol/L [31]. Dong et al. achieved highly efficient dichlorophenols decomposition via photocatalysis using g-C_3_N_4_ [23]. Recent studies have found that mesoporous carbon nitride materials can exhibit excellent catalytic activity through introducing metal atoms inside them. Gianvito Vilé et al. have reported that Cu-based single-atom catalysts developed on a mesoporous carbon nitride carrier exhibited excellent catalytic activity during the synthesis of triazoles [32]. Liu et al. found that the photocatalytic decomposition activity of gemfibrozil can be significantly improved by introducing Ag or Cu atoms into the mesoporous carbon nitride skeleton, which is related to ligand-to-metal charge transfer (LMCT) or ligand-to-metal-to-ligand charge transfer (LMLCT) [33]. Based on the above analysis, g-C_3_N_4_ is hopeful for applications in highly efficient tribocatalytic nitrogen fixation, which is not reported at present.

In this work, the excellent tribocatalytic reduction of nitrogen to ammonia under stirring is achieved in g-C_3_N_4_ which is fabricated via the chemical blowing method. After 10 h stirring at room temperature in the dark, the tribocatalytic ammonia radical (NH_4_^+^) generation rate is up to about 100.56 μmol·L^−1^·g^−1^·h^−1^ using a positive charge scavenger (methanol), which is 3.91 times higher than that without any charge scavengers. In addition, the effects of a negative charge scavenger, the number and speed of stirring rods, and the contact area between the catalyst and stirring rods in tribocatalysis on the performance of g-C_3_N_4_ tribocatalytic N_2_ fixation are also investigated. The possible tribocatalytic mechanism of N_2_ fixation has been also proposed. As the N_2_ fixation research continues to thrive and expand, the finding in this work provides a great potential application to harvest the mechanical energy via tribocatalysis for clean energy production.

## 2. Materials and Methods

### 2.1. Preparation of g-C_3_N_4_ Sample

g-C_3_N_4_ used in this research was prepared according to the reported chemical blowing method [34,35]. A certain amount of ammonium chloride (16 g) and melamine (4 g) were accurately weighed and then mixed together. The mixture was thoroughly ground in a mortar and placed in a crucible. Then, it was stuffed into the muffle furnace, heated from room temperature to 550 °C (6 °C/min), and the calcination time was set to 4 h. After the calcination, the faint yellow product in the crucible was collected and ground through an agate mortar to obtain g-C_3_N_4_.

### 2.2. Characterization

The X-ray diffraction pattern of the g-C_3_N_4_ sample was tested on a D8 Advance diffractometer (Bruker AXS, Karlsruhe, Germany). The micro morphology of the sample was examined with a scanning electron microscope (Gemini SEM 300, ZEISS, Oberkochen, Germany). The chemical properties of the sample were analyzed through using the X-ray photoelectron spectrometer (XPS, ESCALAB 250Xi, Waltham, MA, USA). The infrared spectra of the sample prepared through the KBr tablet pressing method were characterized via the Fourier transform infrared spectrometer (FTIR, Nicolet NEXUS 670, Ramsey, MN, USA). The content of ammonia (NH_4_^+^) was analyzed via a UV-Vis spectrophotometer (Ocean Optics QE65Pro, Dunedin, FL, USA).

### 2.3. Tribocatalytic Performance Measurements

To investigate g-C_3_N_4_ tribocatalytic performance, N_2_ fixation experiments were performed. A total of 50 mL methanol solution (containing 5 mL methanol and 45 mL DI water) mixed with 50 mg g-C_3_N_4_ was contained in a brown bottle. The solution was then placed under shading conditions and stirred at 1000 rpm for 2 h through a polytetrafluoroethylene (PTFE) stirring rod with a specification of ∅ 6 × 20 mm^2^ to achieve adsorption-desorption equilibrium. Then under continuous stirring, the suspension of 3 mL was collected through a rubber-tipped dropper every 2 h of stirring. The supernatan was obtained through a centrifuge (3 min, 6500 rpm). The generation of NH_4_^+^ produced in the process of tribocatalysis was determined through the colorimetric method, and Nessler reagent was selected as the indicator [36,37]. Then 40 μL sodium tartrate solution (0.5 g·mL^−1^) and 60 μL Nessler’s reagent were added dropwise into supernatant and left to stand for 12 min to react sufficiently. Finally, the content of ammonium was analyzed at the peak of ~420 nm via a UV–Vis spectrophotometer.

## 3. Results and Discussion

SEM images of the g-C_3_N_4_ sample before the tribocatalytic N_2_ fixation reaction are depicted in Figure 1. g-C_3_N_4_ samples show the agglomerate morphology composed of many irregular ultra-thin two-dimensional sheet-like structures. From Figure 1, g-C_3_N_4_ catalyst material composed of many huge flakes has a large specific surface area.

The crystal diffraction patterns of g-C_3_N_4_ have been measured with XRD, as shown in Figure 2. The obvious peaks at 2*θ* value of 12.93° and 27.69° are corresponding to the (100) and (002) crystal plane of g-C_3_N_4_ through referring the standard card PDF#87-1526. It is ascribed to the orderly stacking of the conjugated carbon-nitrogen heterocycles in a planar and layered framework, respectively [38,39]. In addition, the high-intensity diffraction peaks and the absence of other impurity peaks indict the excellent synthesis of g-C_3_N_4_.

FTIR spectra depicted in Figure 3 reveal the functional groups of g-C_3_N_4_ before the tribocatalytic N_2_ fixation. The peak around the wave number of 812 cm^−1^ is caused by the out-of-plane bending vibration of the triazine structure [40,41]. The peaks in the wave number range of 1240–1640 cm^−1^ are related to the stretching vibration mode of C–N heterocycle in the g-C_3_N_4_ sample [23]. The broad peaks in the range of 3080–3450 cm^−1^ are ascribed to the stretching vibrations of N−H and O−H groups [39,42].

The elemental states of the g-C_3_N_4_ sample have been performed with XPS measurement, as depicted in Figure 4. From Figure 4a, the measured survey spectrum confirms the inclusion of both C and N elements in g-C_3_N_4_. The high-resolution XPS spectrum of C 1s are presented in Figure 4b. The spectrum of C 1s has two peaks at 284.42 eV and 287.22 eV, which are assign to graphitic carbon adsorbed on the sample surface and sp^2^-bonded carbon in the triazine structure [39]. As shown in Figure 4c, the N 1s spectrum of g-C_3_N_4_ was deconvoluted into three peaks at binding energies of 397.72 eV, 398.87 eV and 400.07 eV, corresponding to C=N−C, N−(C)_3_ and N−H in the sample, respectively [38,43].

To investigate the tribocatalytic activity of the g-C_3_N_4_ sample, N_2_ fixation experiments were performed under stirring. Figure 5 depicts the tribocatalytic N_2_ fixation performance with the different scavengers. When methanol is added as the positive charge scavenger, with the increase of the stirring time, the NH_4_^+^ content increases linearly [44]. The generation rate of NH_4_^+^ can reach 100.56 μmol·L^−1^·g^−1^·h^−1^ after 10 h stirring. It is 3.91 times higher than that without any scavengers. Since the methanol can consume the positive charges, the generation of the negative charges is promoted, which can enhance the tribocatalytic N_2_ fixation performance effectively. It is worth noting that the reductive active radicals, such as the negative charges, are necessary in the N_2_ fixation reaction [45]. TBA can react with hydroxyl radicals (OH) in solution as the radical scavenger [46,47]. Since the formation of ·OH requires the participation of positive charge (q^+^) and hydroxide ion (OH^−^), the addition of TBA promotes the consumption of q+ and effectively prolongs the lifetime of negative charge (q^−^). Therefore, the addition of TBA is beneficial to the tribocatalytic nitrogen fixation of g-C_3_N_4_. Consequently, KBrO_3_ as the negative charge scavenger is used to investigate the important role of the negative charges in this catalytic process [48]. It can be observed that the NH_4_^+^ generation rate is about 0.26 μmol·L^−1^·g^−1^·h^−1^. It can be concluded that the addition of KBrO_3_ is not beneficial for the N_2_ fixation reaction, which accords with the theoretic expectation.

To further investigate the schematic mechanism of the tribocatalytic N_2_ fixation reaction, the control experiments under the different addition have been performed as shown in Figure 6. There is scarcely any generation of NH_4_^+^ without a catalyst, which indicates that the addition of a catalyst is essential for the tribocatalytic N_2_ fixation reaction. Additionally, the tribocatalytic performance associates with the contact area between the g-C_3_N_4_ catalyst and the stirring rod [49]. Therefore, control experiments with the different stirring rods were carried out. Rod I is the commercial PTFE-sealed rod. Two polyvinyl chloride (PVC) electrical tape rings with a width of 1mm were wound on the stirring rod to avoid contact between the catalyst and stirring rod as far as possible, and the stirring rod is named rod II. Obviously, with the decrease of the contact area, the NH_4_^+^ generation rate reduces gradually, indicating that the contact area is an important influencing factor of the tribocatalytic performance.

The influence of the stirring speed on the tribocatalytic performance was also considered. The much faster stirring speed provides much more contact times per minute, that is to say, the contact area is also enlarged relatively per minute. Consequently, the tribocatalytic performance would be enhanced. As depicted in Figure 7, the NH_4_^+^ generation rate is improved significantly from 15.63 to 100.56 μmol·L^−1^·g^−1^·h^−1^ with the stirring speed increasing from 400 to 1000 rpm, and a linear relationship between g-C_3_N_4_ tribocatalytic nitrogen fixation rate and stirring speed is observed. As the stirring speed increases, the friction frequency increases, and more active substances can be produced to participate in the nitrogen fixation process, so an efficient nitrogen fixation can be obtained.

The effect of contact area on the performance of g-C_3_N_4_ tribocatalytic N_2_ fixation is investigated through adjusting the number of stirring rods. Figure 8 shows the nitrogen fixation effect with a different number of stirring rods. Obviously, the total contact area is strongly enlarged as the number of the stirring rods increases. As expected, the NH_4_^+^ generation rate is about 244.02 μmol·L^−1^·g^−1^·h^−1^ using three stirring rods, which is 2.43 times that of only one stirring rod. Typically, the contact area is usually proportional to the number of rods, but the NH_4_^+^ generation rate is not. For instance, the distribution of the catalyst is not uniform in suspension, which leads to inadequate contact between the rods and catalyst, that is to say, each stirring time may not necessarily induce the tribocatalytic reaction. Perhaps there are other influencing factors which influence the tribocatalytic performance. Consequently, the NH_4_^+^ generation rate is not in linear correlation with the number of stirring rods.

Furthermore, the tribocatalytic nitrogen fixation performance of g-C_3_N_4_ was evaluated through comparison with other existing nitrogen fixation research. It can be seen from Table 1 that g-C_3_N_4_ can realize nitrogen fixation through both photocatalysis and tribocatalysis, and it has an excellent performance [50]. In addition, due to its large specific surface area, g-C_3_N_4_ has superior nitrogen fixation activity compared to other materials [51,52,53,54,55].

According to the previous analysis, the tribocatalytic mechanism of g-C_3_N_4_ is drawn in Figure 9. The friction between the g-C_3_N_4_ catalyst and the stirring rod is realized via mechanical stirring, which is accompanied by charge transfer. It is known from the empirical rule of triboelectrification, the g-C_3_N_4_ catalyst is more likely to lose electrons with positive charge than the stirring rod. Therefore, in the process of tribocatalysis, the stirring rod is negatively charged and the catalyst is positively charged. The above process can be represented by Equation (1) [56]:(1)$$g-C_{3}N_{4}+\mathrm{stirring}\;\mathrm{rod}\overset{\mathrm{Friction}}{\to }g-C_{3}N_{4}\left(q^{+}\right)+\mathrm{stirring}\;\mathrm{rod}\left(q^{-}\right)$$

Then methanol as the positive charge scavenger will react with q+ to avoid NH_4_^+^ being oxidized [37]. In addition, q^−^ will react with N_2_ to produce NH_4_^+^, which is dissolved in DI water to form NH_4_^+^. The main reaction process is described in Equations (2)–(4) [57,58]:(2)$$q^{+}+{\mathrm{CH}}_{3}\mathrm{OH}\to {\mathrm{CH}}_{3}{\mathrm{OH}}^{+}$$
(3)$$6{\;q}^{-}+N_{2}+6H^{+}\to 2{\mathrm{NH}}_{3}$$
(4)$${\mathrm{NH}}_{3}+H_{2}O\to {\mathrm{NH}}_{3}\cdot H_{2}O⇌{\mathrm{NH}}_{4}{}^{+}+{\mathrm{OH}}^{-}$$

In this work, g-C_3_N_4_ exhibits excellent tribocatalytic performance in the N_2_ fixation process. Though, as a matter of fact, the applications of the tribocatalysis are not restricted in this field. In the past few years, He et al. have achieved excellent tribocatalytic performance on removal of the heavy metal ion Cr^6+^ through using commercial iron turning with amorphous iron oxides. After 36 h stirring, the removal ratio of Cr^6+^ can reach about 98% [59]. Additionally, Li et al. have successfully produced flammable gases such as CO, CH_4_ and H_2_ through harvesting mechanical energy with TiO_2_ nanoparticles [20]. In the future, g-C_3_N_4_ has potential applications in wastewater treatment and energy generation fields via tribocatalysis.

## 4. Conclusions

In summary, g-C_3_N_4_ has been fabricated successfully via the chemical blowing method and shows excellent tribocatalytic performance in the reduction of nitrogen to NH_4_^+^. After 10 h of continuous stirring at 1000 rpm in the dark, the generation rate of NH_4_^+^ can reach 100.56 μmol·L^−1^·g^−1^·h^−1^ using methanol as a positive charges scavenger, which is 3.91 times higher than that without any scavengers. Furthermore, the performance of the tribocatalytic nitrogen fixation of g-C_3_N_4_ can be effectively optimized through increasing the stirring speed or number of stirring rods. Consequently, g-C_3_N_4_ has the remarkable potential application in the tribocatalytic N_2_ fixation reaction. Tribocatalysis has a bright application prospect in energy development fields such as nitrogen fixation, carbon dioxide reduction and water decomposition in the future.

## Figures and Tables

**Figure 1 nanomaterials-12-01981-f001:**
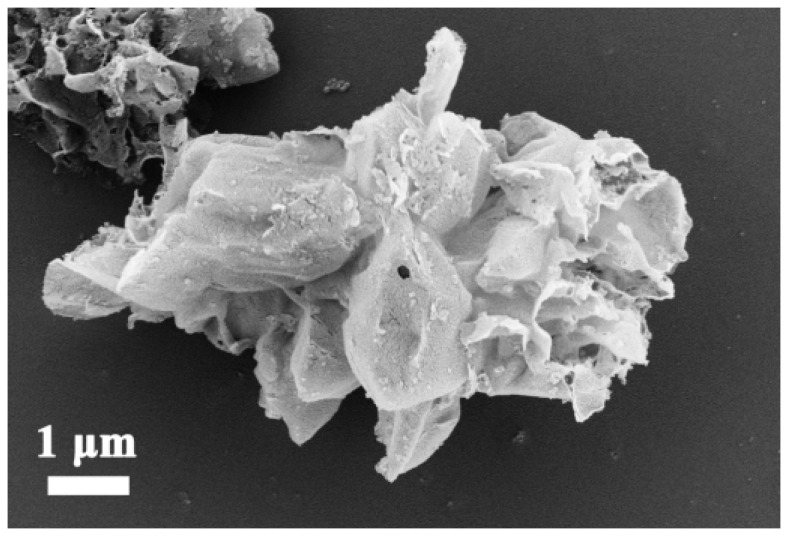
SEM image of g-C_3_N_4_.

**Figure 2 nanomaterials-12-01981-f002:**
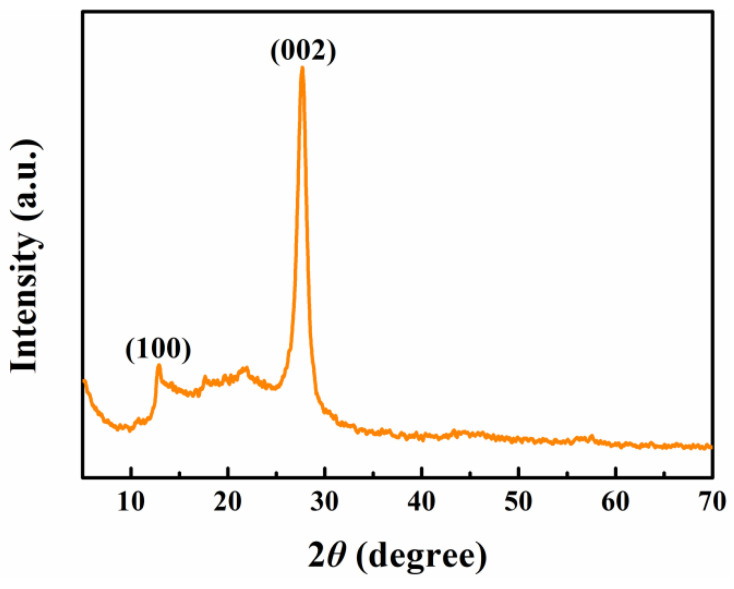
XRD patterns of g-C_3_N_4_.

**Figure 3 nanomaterials-12-01981-f003:**
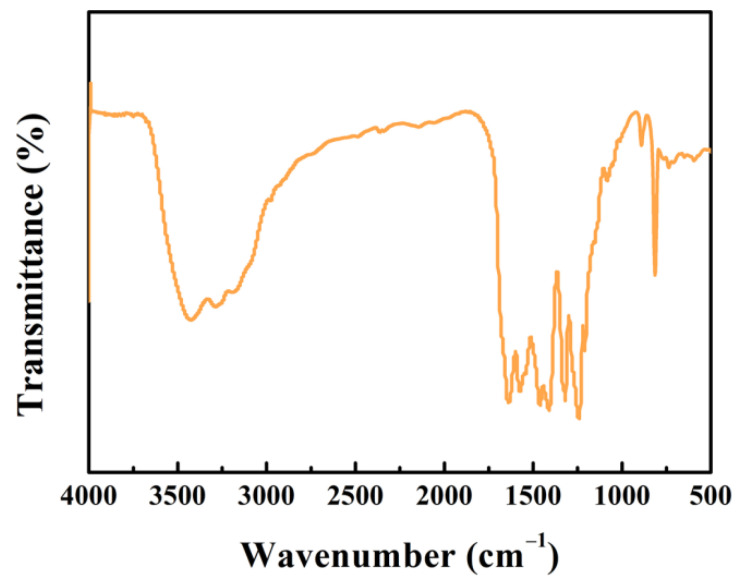
FTIR spectra of g-C_3_N_4_.

**Figure 4 nanomaterials-12-01981-f004:**
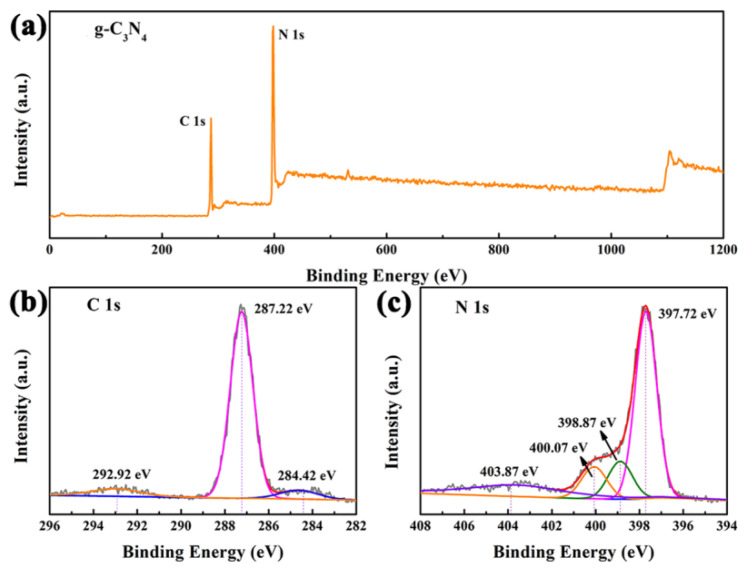
XPS spectra of g-C_3_N_4_ sample: (**a**) survey, (**b**) C 1s, and (**c**) N 1s spectra.

**Figure 5 nanomaterials-12-01981-f005:**
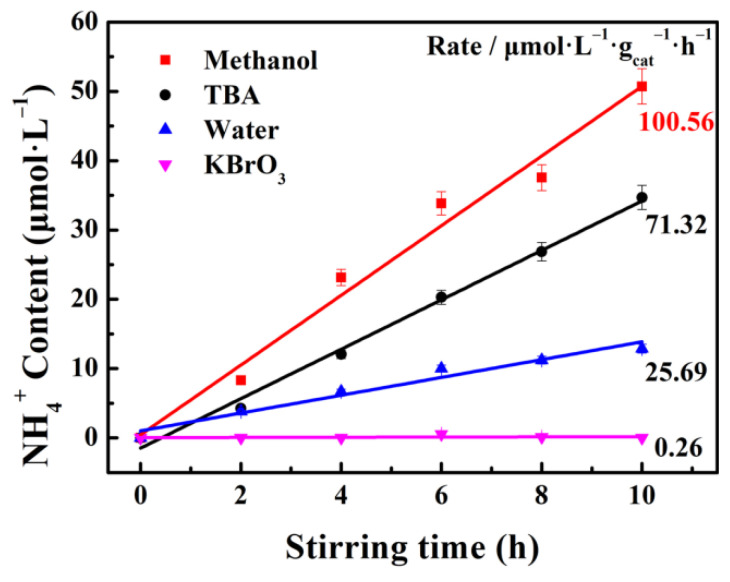
Tribocatalytic N_2_ fixation performance of g-C_3_N_4_ with the different scavengers.

**Figure 6 nanomaterials-12-01981-f006:**
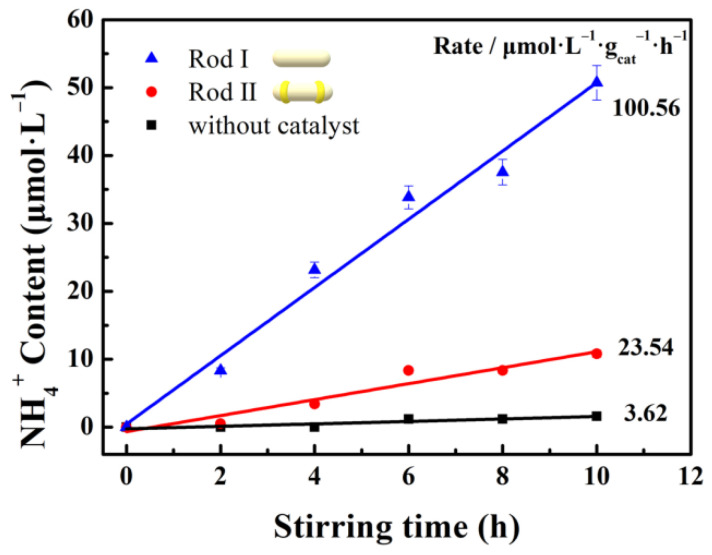
Tribocatalytic N_2_ fixation performance of g-C_3_N_4_ with the different kinds of rods or without catalyst.

**Figure 7 nanomaterials-12-01981-f007:**
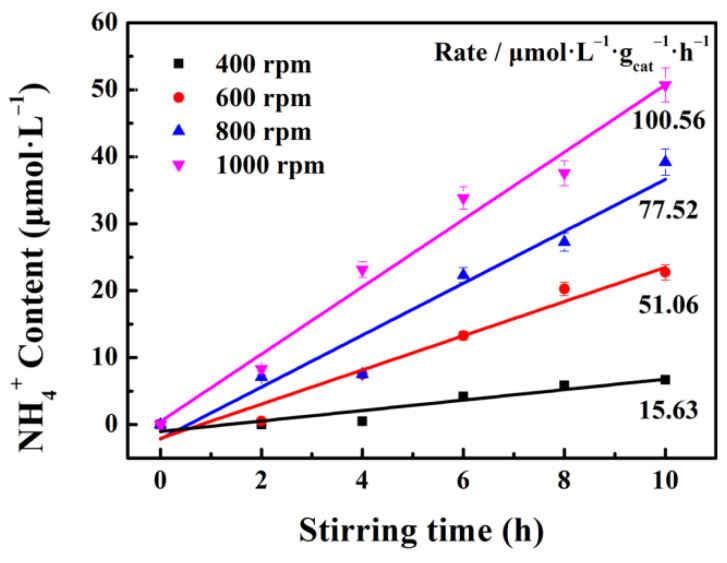
Tribocatalytic N_2_ fixation performance of g-C_3_N_4_ under the different stirring speed.

**Figure 8 nanomaterials-12-01981-f008:**
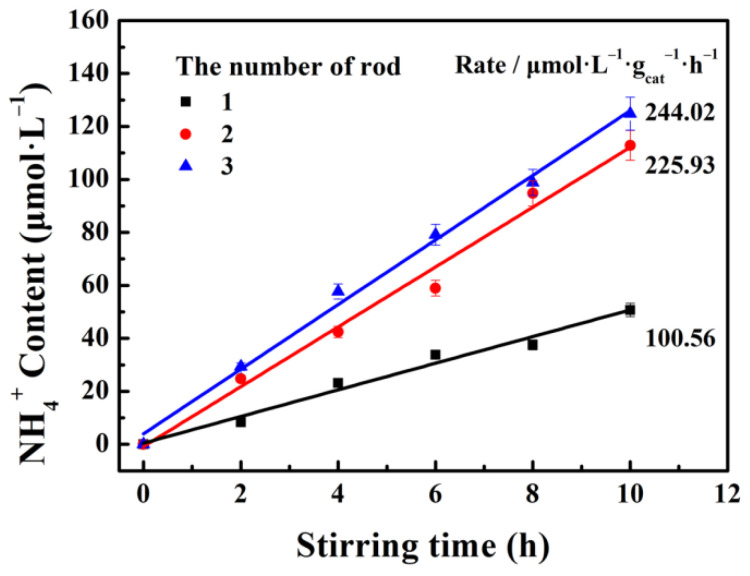
Tribocatalytic N_2_ fixation performance of g-C_3_N_4_ with the different number of stirring rods.

**Figure 9 nanomaterials-12-01981-f009:**
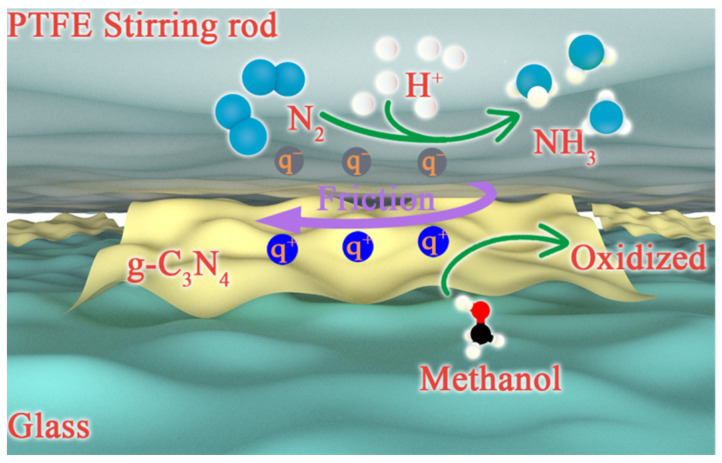
The schematic diagram for the tribocatalytic mechanism of g-C_3_N_4_.

**Table 1 nanomaterials-12-01981-t001:** Summary of ammonia fixation performance of different catalysts and different catalytic methods.

Catalysts	Ammonia Generation Rate/μmol·L^−1^·g^−1^·h^−1^	Nitrogen Source	Scavenger	Catalytic Method
g-C_3_N_4_	100.56	air	methanol	Tribocatalysis [this work]
g-C_3_N_4_	160	air	methanol	Photocatalysis [50]
P25	52	N_2_	water	Photocatalysis [51]
BiOCl	68.9	N_2_	methanol	Photocatalysis [52]
FeS_2_/CeO_2_	90	N_2_	water	Photocatalysis [53]
KTa_0.5_Nb_0.5_O_3_	13.2	air	methanol	Piezocatalysis [54]
Ag/Bi_5_O_7_I	65.4	air	water	Piezocatalysis [55]

## Data Availability

The data presented in this study are available in this article.

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
