# Peer review of "Strong Tribocatalytic Nitrogen Fixation of Graphite Carbon Nitride g-C3N4 through Harvesting Friction Energy"

_nanomaterials, 2022, doi:10.3390/nano12121981_

Round 1

Reviewer 1 Report

The author studied the performance of catalytic reaction for ammonia generation using 2D g-C3N4 and reaction mechanism through kinetic analysis. The analysis of various reaction conditions and reaction mechanisms appeals to the value of this paper. However, the following major revisions are required as the quality of this study is deteriorated due to its inexperienced composition and interpretation.

1) The author begins the discussion with characterization before and after the catalytic reaction, and discusses the catalytic reaction results later. It is doubtful whether this contents order will help to understand the overall context, compared to the usual description of characteristics evaluation, reactivity evaluation, and stability evaluation in the order of evaluation.

The content of the characteristic evaluation itself is also poor, and the fact that there is no significant difference before and after the reaction is also insufficient as a content that should be mainly discussed. If there is no difference, one way is to state that there was no significant difference in characteristics after the catalytic reaction analysis and move it to the supporting information. For example, this order can be applied: characterization pristine 2D g-C3N4, reaction analysis, mechanism discussion + stability test if possible (including characterization).

Or, if the author wants to stick to the original style, 1) it is strongly recommended to also view the TEM image to make sure that the g-C3N4 is 2D structure and is not different before and after the reaction. 2) It is good to show the XPS results before and after the reaction as a set. 3) If possible, it is recommended that changes in specific surface area are also observed. Moreover, it is necessary to clearly state under what conditions the samples are being compared.

2) 3 Results →  3 Results and Discussion

  4 Discussion →  4. Conclusion or Summary

3) It is necessary to compare the performance of the synthesized material with other references.

4) Experimental Method: What are the N2 and H+ sources? Have you loaded N2 and hydrogen separately?

5) Figure 5: MeOH vs TBA, Both of them can capture q+, then, why the author compared them? And, make sure that subsequent experiments were conducted using MeOH.

6) Figure 6, Explain the reason for this experiment in more detail for readers. Why this experiment is necessary?

7) Figure 7, 8: What about the change in reaction temperature with increasing RPM or increasing the number of bars? Can it be ignored?

8) Equation 1 is ambiguous. Be sure to indicate that the catalyst is + and the rod is - charged.

9)  Equation 2,  q+ + CH3OH → CH3OH+ is better.

10) What is the meaning of NH3・H2O in Eq 4.? Is it an appropriate notation? Or is it a necessary step?

11) Figure 9: Green is Glass, does it mean beaker? Is it important in the reaction mechanism? Are there no water molecules?

12) line 230-238, this content can be moved to introduction.

Author Response

Response to Reviewer 1 Comments

Point 1: The author begins the discussion with characterization before and after the catalytic reaction, and discusses the catalytic reaction results later. It is doubtful whether this contents order will help to understand the overall context, compared to the usual description of characteristics evaluation, reactivity evaluation, and stability evaluation in the order of evaluation.

The content of the characteristic evaluation itself is also poor, and the fact that there is no significant difference before and after the reaction is also insufficient as a content that should be mainly discussed. If there is no difference, one way is to state that there was no significant difference in characteristics after the catalytic reaction analysis and move it to the supporting information. For example, this order can be applied: characterization pristine 2D g-C3N4, reaction analysis, mechanism discussion + stability test if possible (including characterization).

Or, if the author wants to stick to the original style, 1) it is strongly recommended to also view the TEM image to make sure that the g-C3N4 is 2D structure and is not different before and after the reaction. 2) It is good to show the XPS results before and after the reaction as a set. 3) If possible, it is recommended that changes in specific surface area are also observed. Moreover, it is necessary to clearly state under what conditions the samples are being compared.

Response: Thanks for reviewer’s suggestion, we have adjusted the sequence of text on basis of the reviewer’s suggestion.

Point 2: The 3 Results →  3 Results and Discussion

        4 Discussion →  4. Conclusion or Summary

Response: Thanks for reviewer’s suggestion, the suggestion modification has been done.

Point 3: It is necessary to compare the performance of the synthesized material with other references.

Response 3: Thanks for your suggestion, we have compared at least 10 references of nitrogen fixation using other synthesized materials via other catalytic technology, as shown in Table 1.

Table I. The comparison of the rate of ammonia formation performance of different catalyst types and different catalytic methods

Catalysts

ammonia generation rate/μmol·L−1·g−1·h−1

nitrogen source

scavenger

Catalytic method

g-C3N4

100.56

air

methanol

Tribocatalysis [this work]

g-C3N4

160

air

methanol

Photocatalysis [Reference]

P25

52

N2

water

Photocatalysis [Reference]

 BiOCl

68.9

N2

methanol

Photocatalysis [Reference]

 FeS2/CeO2

90

N2

water

Photocatalysis [Reference]

KTa0.5Nb0.5O3

13.2

air

methanol

Piezocatalysis [Reference]

Ag/Bi5O7I

65.4

air

water

Piezocatalysis [Reference]

Point 4: Experimental Method: What are the N2 and H+ sources? Have you loaded N2 and hydrogen separately?

Response 4: Thanks for reviewer’s suggestions. It’s well known that >70% content of air is N2. For the need of future’s practical application, in our experiment, the N2 is directly from the natural air, and the H+ is from reaction solution.

Point 5: Figure 5: MeOH vs TBA, Both of them can capture q+, then, why the author compared them? And, make sure that subsequent experiments were conducted using MeOH.

Response 5: Thanks for reviewer’s suggestion. According to the reference, TBA used in our experiment can capture ·OH active species, which may play an important role in tribocatalytic nitrogen fixation process. In our other experiment, only MeOH are used to capture q+ induced by triboelectricty.

Point 6: Figure 6, Explain the reason for this experiment in more detail for readers. Why this experiment is necessary?

Response 6: Thanks for reviewer’s suggestion, the analysis on fig. 6 has been carefully revised. The final text related on Fig.6 is list as following:  

To further investigate the schematic mechanism of the tribo-catalytic N2 fixation reaction, the control experiments under the different addition have been performed as shown in Fig. 6. There is scarcely any generation of NH4+ without catalyst, which indicates that the addition of catalyst is essential for the tribo-catalytic N2 fixation reaction. Additionally, the tribocatalytic performance associates with the contact area between g-C3N4 catalyst and the stirring rod. Therefore, the control experiments with the different stirring rods have been carried out. Rod Ⅰ is the commercial PTFE-sealed rod. Two polyvinyl chloride (PVC) electrical tape rings with a width of 1 mm are wound on the stirring rod to avoid the contact between catalyst and stirring rod as far as possible, and the stirring rod is named rod Ⅱ. Obviously, with the decrease of the contact area, the NH4+ generation rate reduces gradually, indicating that the contact area is an important influencing factor of the tribo-catalytic performance.

In order to investigate the influence of catalyst and contact between the stirring rod and catalyst, this control experiment is also performed in our experiment.

Point 7: Figure 7, 8: What about the change in reaction temperature with increasing RPM or increasing the number of bars? Can it be ignored?

Response 7: Thanks for reviewer’s suggestion. In the tribocatalytic process, with the increasing of RPM or the numbers of stirring bars, the temperature of solution would increase. In theory, the slight temperature fluctuation <5 oC does not affect our experimental result.

Point 8: Equation 1 is ambiguous. Be sure to indicate that the catalyst is + and the rod is - charged.

Response 8: Thanks for reviewer’s suggestion, the equation 1 has been revised as following:

g-C3N4 + stirring rod→g-C3N4 (q+) + stirring rod (q-)

Point 9: Equation 2,  q+ + CH3OH → CH3OH+ is better.

Response 9: Thanks for reviewer’s suggestion, the equation 2 has been revised.

Point 10: What is the meaning of NH3·H2O in Eq 4.? Is it an appropriate notation? Or is it a necessary step?

Response 10: Thanks for reviewer’s suggestion. For equation 4, the produced NH3 in the tribocatalytic reaction process would be dissolved in solution to generate NH4+ , which can be effectively detected for the calculation of the NH4+ or NH3 content.

Point 11: Figure 9: Green is Glass, does it mean beaker? Is it important in the reaction mechanism? Are there no water molecules?

Response 11: Thanks for reviewer’s suggestion. The green is glass, which means beaker. In the tribo-catalytic reaction, the role of beaker is fixing catalyst in order to rube more effectively between the stirring rod and catalyst. Since H+ is from water molecules, and water molecules don’t participate in the catalytic reaction directly, hence there are no water molecules in figure 9.

Point 12: line 230-238, this content can be moved to introduction.

Response 12: Thanks for reviewer’s suggestion. In the last paragraph of the “Introduction”, I have summarized the main work content and simple conclusion of our tribocatalytic N2 fixation to attract the audience to continue to read down, which obey the ordinary writing custom of academic research article.

Reviewer 2 Report

The article describes a new direction in catalysis - tribocatalysis - for an unusually promising process - nitrogen fixation into ammonia. Although the research seems very interesting to me, a number of questions arise:

1. What is the yield of the target product in the chemical blow method of synthesis of graphite-like carbon nitride.

2. HRTEM with mapping should be provided in order to elucidate the structure of the photocatalyst.

3. For all kinetic curves, the error of measurements should be shown.

4. g-C3N4 is known to be an active photocatalyst. Why don't the authors compare the photocatalytic and tribocatalytic reduction of the nitrogen molecule.

5. A table should be provided comparing numerical results on the rate of ammonia formation with published data on this process, including photocatalytic nitrogen reduction.

Author Response

Response to Reviewer 2 Comments

Point 1: What is the yield of the target product in the chemical blow method of synthesis of graphite-like carbon nitride.

Response 1: In this study, through the chemical blowing method, 4 g of melamine and 16 g of ammonium chloride can be calcined at high temperature to produce about 1.6 g of g-C3N4 with a yield of about 40 %.

Point 2: HRTEM with mapping should be provided in order to elucidate the structure of the photocatalyst.

Response 2: Thanks for the reviewer’s helpful suggestion. The micromorphology and crystal structure of the g-C3N4 catalyst were tested through using SEM and XRD in our experiment. Due to the COVID-19, there is no condition for us to do the HRTEM.

Point 3: For all kinetic curves, the error of measurements should be shown.

Response 3: Thanks for the reviewer’s comments. the errors of measurements have been added to all experimental data points.

Point 4: g-C3N4 is known to be an active photocatalyst. Why don't the authors compare the photocatalytic and tribocatalytic reduction of the nitrogen molecule.

Response 4:

Table I. Photocatalytic and tribocatalytic nitrogen reduction of g-C3N4.

Catalysts

ammonia generation rate/μmol·L−1·g−1·h−1

nitrogen source

scavenger

Catalytic method

g-C3N4

100.56

air

methanol

Tribocatalysis [This work]

g-C3N4

160

air

methanol

Photocatalysis [Reference]

W18O49/g-C3N4

144

N2

ethanol

Photocatalysis [Reference]

Thanks for the reviewer’s helpful comments. At present, there have been many reports on the photocatalytic nitrogen reduction of g-C3N4. Its photocatalytic nitrogen reduction performance has been further optimized through defect introduction or heterostructure construction. Therefore, it is of great significance to compare the difference between the photocatalytic and tribocatalytic nitrogen reduction of g-C3N4. Therefore, I have made a comparison through the following table I. As shown in table I, the tribocatalytic N2 fixation in our work is comparable to the widely-reported phtocatalytic N2 fixation.

Point 5: A table should be provided comparing numerical results on the rate of ammonia formation with published data on this process, including photocatalytic nitrogen reduction.

Response 5: Thanks for the reviewer’s comments. We have added the following table to make a comparison with numerical results on the rate of ammonia formation with published data on this process.

Table I. The comparison of the rate of ammonia formation performance of different catalyst types and different catalytic methods

Catalysts

ammonia generation rate/μmol·L−1·g−1·h−1

nitrogen source

scavenger

Catalytic method

g-C3N4

100.56

air

methanol

Tribocatalysis [this work]

g-C3N4

160

air

methanol

Photocatalysis [Reference]

P25

52

N2

water

Photocatalysis [Reference]

 BiOCl

68.9

N2

methanol

Photocatalysis [Reference]

 FeS2/CeO2

90

N2

water

Photocatalysis [Reference]

KTa0.5Nb0.5O3

13.2

air

methanol

Piezocatalysis [Reference]

Ag/Bi5O7I

65.4

air

water

Piezocatalysis [Reference]

Round 2

Reviewer 1 Report

The authors provided Table 1 in the author's answers. But, there is no Table 1 and related content in the main manuscripts. Please double check it.

and section 4 should be 4. Conclusion  (or 4. Summary). I recommned 4. Conclusion 

Author Response

Response to Reviewer 1 Comments

Point 1: The authors provided Table 1 in the author's answers. But, there is no Table 1 and related content in the main manuscripts. Please double check it.

Response: Thanks for reviewer’s helpful suggestion. Table 1 and some related discussions have been put into the main manuscript.

Furthermore, the tribocatalytic nitrogen fixation performance of g-C3N4 is evaluated through comparing with other existing nitrogen fixation research. It can be seen from table 1 that g-C3N4 can realize nitrogen fixation through both photocatalysis and tribocatalysis, and has a excellent performance [50]. In addition, due to its large specific surface area, g-C3N4 has a superior nitrogen fixation activity than other materials [51–55].

Table 1. Summary of ammonia fixation performance of different catalysts and different catalytic methods.

Catalysts

Ammonia Generation Rate/μmol·L−1·g−1·h−1

Nitrogen Source

Scavenger

Catalytic Method

g-C3N4

100.56

air

methanol

Tribocatalysis [this work]

g-C3N4

160

air

methanol

Photocatalysis [50]

P25

52

N2

water

Photocatalysis [51]

BiOCl

68.9

N2

methanol

Photocatalysis [52]

FeS2/CeO2

90

N2

water

Photocatalysis [53]

KTa0.5Nb0.5O3

13.2

air

methanol

Piezocatalysis [54]

Ag/Bi5O7I

65.4

air

water

Piezocatalysis [55]

Point 2: section 4 should be 4. Conclusion  (or 4. Summary). I recommned 4. Conclusion 

Response: Thanks for reviewer’s suggestion, the suggestion modification has been done.
